# Expression in *Pichia pastoris* of Thermostable Endo-1,4-β-xylanase from the Actinobacterium *Nocardiopsis halotolerans*: Properties and Use for Saccharification of Xylan-Containing Products

**DOI:** 10.3390/ijms25169121

**Published:** 2024-08-22

**Authors:** Alexander V. Lisov, Oksana V. Belova, Andrey A. Belov, Zoya A. Lisova, Alexey S. Nagel, Andrey M. Shadrin, Zhanna I. Andreeva-Kovalevskaya, Maxim O. Nagornykh, Marina V. Zakharova, Alexey A. Leontievsky

**Affiliations:** 1Federal Research Center “Pushchino Scientific Center for Biological Research of the Russian Academy of Sciences”, G.K. Skryabin Institute of Biochemistry and Physiology of Microorganisms, Russian Academy of Sciences, 142290 Pushchino, Russia; 2Faculty of Soil Science, Lomonosov Moscow State University, 119991 Moscow, Russia

**Keywords:** endo-1,4-β-xylanase, heterologous expression, *Pichia pastoris* fermentation, *Nocardiopsis halotolerans*

## Abstract

A gene encoding a polysaccharide-degrading enzyme was cloned from the genome of the bacterium *Nocardiopsis halotolerans*. Analysis of the amino acid sequence of the protein showed the presence of the catalytic domain of the endo-1,4-β-xylanases of the GH11 family. The gene was amplified by PCR and ligated into the pPic9m vector. A recombinant producer based on *Pichia pastoria* was obtained. The production of the enzyme, which we called NhX1, was carried out in a 10 L fermenter. Enzyme production was 10.4 g/L with an activity of 927 U/mL. Purification of NhX1 was carried out using Ni-NTA affinity chromatography. The purified enzyme catalyzed the hydrolysis of xylan but not other polysaccharides. Endo-1,4-β-xylanase NhX1 showed maximum activity and stability at pH 6.0–7.0. The enzyme showed high thermal stability, remaining active at 90 °C for 20 min. With beechwood xylan, the enzyme showed K_m_ 2.16 mg/mL and V_max_ 96.3 U/mg. The products of xylan hydrolysis under the action of NhX1 were xylobiose, xylotriose, xylopentaose, and xylohexaose. Endo-1,4-β-xylanase NhX1 effectively saccharified xylan-containing products used for the production of animal feed. The xylanase described herein is a thermostable enzyme with biotechnological potential produced in large quantities by *P. pastoria*.

## 1. Introduction

Xylan is a polysaccharide widely distributed in plant cell walls and is one of the main components of hemicellulose. The xylan polymer chain consists of xylose residues linked by a β-1,4-O-glycosidic bond and arabinose and/or glucuronic acid residues linked to the xylan backbone by a 2,3 bond [1]. Residues of acetic and ferulic acids can also be attached to xylose via an ester bond. Xylan is utilized by many bacteria and fungi. For this purpose, microorganisms produce various enzymes, including those capable of cleaving off side groups of xylan, but the decomposition of the main chain occurs under the action of hydrolase (endo1,4-βxylanase (EC 3.2.1.8)) and lytic polysaccharide monooxygenases [2]. Endo-1,4-β-xylanase catalyzes the hydrolysis of the glycosidic bond between xylose residues; the enzyme is widely distributed among living organisms [3]. Based on structural, genomic and functional data, endo-1,4-β-xylanase from various sources is classified into different classes of glycoside hydrolases (GH), which is reflected in the CAZy database (carbohydrate-active enzymes, http://www.cazy.org/ accessed on 27 June 2024) [4]. Proteins with endo-1,4-β-xylanase activity are found in various GH families, but the most studied enzymes are from the GH10 and GH11 families.

Endo-1,4-β-xylanase is one of the most widely used enzymes in practice. The high interest of researchers in endo-1,4-β-xylanase and the search for new sources of the enzyme with new properties are due to the practical significance of the enzyme. This enzyme is used to bleach paper pulp; improve the nutritional properties of animal feed; and improve the properties of dough, clarify juices, and produce probiotics in the food industry [5,6,7]. One of the important practical properties of enzymes, expanding the possibility of their practical use, is their thermal stability [8]. The use of heat-stable endo-1,4-β-xylanase allows it to be used at elevated temperatures in animal feed pelleting and paper pulp bleaching [9,10]. Endo-1,4-β-xylanase is obtained from various producers, both natural and recombinant [11,12]. Thermostable endo-1,4-β-xylanases are obtained primarily from thermophilic microorganisms [8], as well as by using protein engineering methods [13].

This work describes the preparation and study of the properties of endo-1,4-β-xylanase from the actinobacterium *Nocardiopsis halotolerans*. *N. halotolerans* was isolated from saline soils [14]. A complete genome has been determined for this bacterium and deposited in GenBank. The *N. halotolerans* genome contains genes encoding glycosyl hydrolases, including endo-1,4-β-xylanases. The goal of the work was to obtain a preparation of recombinant endo-1,4-β-xylanase from the actinobacterium *N. halotolerans*, produce it in recombinant *Pichia pastoris* and study its properties and biotechnological potential.

## 2. Results and Discussion

### 2.1. Characteristics of the Amino Acid Sequence and Phylogenetic Analysis

The genome of *N. halotolerans* contains genes potentially encoding polysaccharide degrading enzymes, including the gene encoding the protein WP_017571782.1. Analysis of the amino acid sequence of the protein using the InterPro program revealed the presence of a signal peptide within the range of 1–42 amino acid residues. This indicates possible extracellular secretion of the protein, which is common for xylanases from different sources [15]. Amino acid residues 43–229 constitute the catalytic domain of the GH11 family of glucohydrolases (http://www.cazy.org/ accessed on 27 June 2024), characteristic of xylanases. Amino acid residues 241–333 form the polysaccharide binding domain of the CBM2 family (http://www.cazy.org/CBM2.html accessed on 27 June 2024), which is also found in xylanases [16]. The sequence of the endo-1,4-β xylanase gene from the bacterium *N. halotolerans*, obtained by PCR, fully corresponded to the sequence of the gene deposited in GenBank.

A search in GenBank showed the presence of a large number of homologous proteins. Twenty-eight proteins had at least 80% homology and their amino acid sequences were aligned (Figure 1). The closest homology of 85.9% was with the protein from *Nocardiopsis* sp. FR6. Each of the proteins contains a TAT signal peptide, a GH11 catalytic domain, and a CBM2 polysaccharide binding domain. Identical amino acids are located mainly in GH11 and CBM2. Actinobacteria *Actinomadura* sp. 6K520 contains a single amino acid insertion in the GH11 domain. Deletions and insertions of amino acids occur mainly in the TAT signal peptide and interdomain linker. Phylogenetic analysis showed that proteins within genus *Nocardiopsis* are closer to each other (Figure 2), except for the protein from *Glycomycesfuscus*, which showed high similarity to the proteins from *Nocardiopsis*. At the same time, the protein from *Nocardiopsis deserti* showed similarity with proteins from genera *Actinomadura* and *Nonomuraea*, which form a separate group. The homologous proteins have not been studied previously, so it cannot be said that their properties are similar to endo-1,4-β-xylanase from *N. halotolerans*.

### 2.2. Cloning, Expression and Purification

The gene encoding the protein WP_017571782.1, which we named NhX1, was amplified by PCR. Next, the amplified gene encoding NhX1 was inserted into the expression vector pPic9m. Using this vector, a recombinant producer of *P. pastoris* was obtained. NhX1 protein expression was carried out in a fermenter. Recombinant proteins in the *P. pastoris*/AOX1 promoter system can be produced using several strategies: accumulation of a high concentration of biomass followed by protein induction with methanol; co-cultivation of the producer with methanol and a substrate that does not inhibit the AOX1 promoter; and growing the producer on methanol [17]. To produce NhX1, we used the strategy of accumulating a high concentration of producer biomass. Therefore, a “rich” medium containing a large amount of nutritional components was used. Glycerol was used as the main carbon source. This substance represses the AOX1 promoter [18], but methanol for induction was added after the glycerol was depleted in the medium. Glycerol depletion was monitored by a decrease in the respiration of the producer culture. The *P. pastoris* culture transitioned from exponential growth to the stationary phase by 20 h of growth (Figure 3B). By the 25th hour of growth, the glycerol was depleted and respiration of the *P. pastoris* yeast decreased, after which methanol was added to the medium in an amount of 1 mL per liter of medium to induce methanol metabolism in the *P. pastoris* culture. Two hours later, after the first portion of methanol was disposed of, a second portion was added in the same amount. At the 29th hour of growth of the *P. pastoris* culture, a continuous supply of methanol was turned on in an amount of 2 mL per hour per liter of medium. The methanol supply rate was increased over three hours to 5 mL per hour per liter of medium. A further increase in the supply of methanol was not carried out, since it inhibited the respiration of the *P. pastoris* culture.

Cultivation of *P. pastoris* was carried out until 90 h of growth, after which the respiration of the culture began to decrease. Protein production was observed during methanol induction (Figure 3A). By the end of cultivation, the protein concentration was 10.4 g/L, and the activity of endo-1,4-β-xylanase NhX1 was 927 U/mL. 

The expression system based on *P. pastoris* is widely used for the production of proteins, including the production of endo-1,4-β-xylanases [19,20]. To obtain a large amount of xylanase, the approach of accumulating a large concentration of biomass during fermentation in a bioreactor was used previously. For example, recombinant xylanase from the bacterium *Fibrobacter succinogenes* was obtained in an amount of 7752 U/mL culture [20]. Optimization of xylanase production from the fungus *Myceliophthora thermophila* resulted in an activity of 2503 U/mL [21].

### 2.3. Properties of NhX1

The NhX1 protein was purified to an electrophoretically homogeneous state (Figure 3C). Its properties were then studied. The enzyme catalyzed the hydrolysis of xylan with a specific activity of 89 U/mg. The NhX1 enzyme did not catalyze the hydrolysis of amorphous cellulose, CM-cellulose, beta-glucan, lichenin, or starch. The optimal pH value for xylan hydrolysis was in the neutral region and amounted to 6.0–7.0 (Figure 4A). At the same time, in the acidic (less than 3.0) and alkaline (more than 9.0) pH ranges, NhX1 activity decreased and was less than 50% of the maximum. The NhX1 enzyme was more stable in the neutral, slightly acidic and slightly alkaline pH ranges (Figure 4B). Neutral xylanases were previously found in actinobacteria [22]. Xylanases with alkaline and acidic pH optimums are also known [23,24]. 

Xylanase NhX1 had a temperature optimum of about 80–90 °C. The enzyme exhibited high thermal stability. Xylanase NhX1 retained 50 and 40% activity, respectively, for an hour at 60 and 70 °C with a half-inactivation time (t ½) of 57 and 31 min, respectively (Figure 4D); at 80 °C it was completely inactivated within an hour (t ½ 14 min); and at 90 °C the enzyme completely lost activity within 20 min (t ½ 4 min). Thermostable xylanases have been described previously. Basically, these xylanases have been obtained from thermophilic and hyperthermophilic microorganisms. Xylanase from the thermophilic bacterium *Caldicoprobacter algeriensis* was stable for more than an hour even at 100 °C [25]. The mutant xylanase of the hyperthermophile *Thermotoga maritima* also retained activity at 100 °C [26]. However, thermostable enzymes have not always been obtained from thermophilic microorganisms. A xylanase stable at 90 °C for several hours was obtained from the psychrotolerant bacterium *Planomicrobium glaciei* [27]. The actinobacterium *N. halotolerans* is mesophilic and its growth is not observed above 40 °C [14]. However, this actinobacterium was isolated from the saline soil of Kuwait, a country with a hot climate. It is possible that the presence of thermostable xylanase in *N. halotolerans* is associated with the operation of this enzyme under conditions of local soil heating, which allows the actinobacterium to survive unfavorable conditions. 

The kinetic constants of NhX1 to beechwood xylan were determined (Appendix A). The value of K_m_ was 2.16 ± 0.1 mg/mL, and the value of V_max_ was 96.3 ± 3.5 U/mg. This is consistent with the literature data. Thus, the K_m_ values of xylanases from various sources vary within 0.15–49.5 mg/mL, and the V_max_ within 0.017–7610 U/mg [6,28].

The products of xylan hydrolysis under the action of NhX1 were mainly xylo-oligosaccharides—xylobiose, xylotriose, xylopentaose, and xylohexaose (Figure 5). The formation of xylotetraose was observed, but it almost completely disappeared by 24 h of reaction. Also, by the 24th hour of the reaction, a small amount of xylose was formed. The formation of such products under the action of NhX1 is consistent with data obtained for GH11 family xylanases from other sources. GH11 xylanases are characterized by hydrolysis of linear regions of xylan molecules with the formation of xylo-oligosaccharides, which are then weakly transformed [29]. Unlike GH10 family xylanases, GH11 xylanases are not characterized by xylose formation [30]. However, there are xylanases of the GH11 family that form xylose from xylan in small quantities [31]. It has also been shown that GH11 xylanases are capable of utilizing xylo-oligosaccharides, including xylotetraose [32].

Xylanase is used for saccharification of plant materials. Saccharification using xylanase is used to produce bioethanol [33] and in food industry [34]. At the same time, to increase saccharification, various techniques are used, such as effects on plant biomass temperature and chemicals [35,36] and the treatment of biomass with a mixture of xylanase and other enzymes [37,38]. Plant biomass is a complex material containing components of various natures, for example, phenolic compounds. To reduce the negative effect of phenolic compounds on saccharification, another enzyme, such as laccase, can be used [39]. 

Xylanase is also actively used to improve the nutritional properties of animal feed. This enzyme degrades xylans in plant-based feeds and thus reduces the chyme viscosity and increases intestinal nutrient absorption [40]. The effectiveness of xylanase supplementation in increasing weight gain in poultry has been well studied, although the effect is more moderate in pigs. [41]. Xylanase NhX1 effectively saccharified common plant xylan-containing products used in animal feed—rye, wheat, oats, barley, and wheat bran (Figure 6). The formation of reducing sugars occurred with different efficiencies for different raw materials in concentrations of 1, 5, and 10%. The concentration of reducing sugars increased with increasing substrate concentration for rye, barley, and wheat bran. The amount of reducing sugars at substrate concentration 10% increased by 32% and 38% in rye and barley, respectively, and in wheat bran the increase in reducing sugars at this concentration was 34%. In wheat, the increase in the concentration of reducing sugars at 5% and 10% of substrate was the same—39%. The difference in the efficiency of the formation of reducing sugars may be due to the fact that xylanes are present in different concentrations in different cereals [42]. Also, in various plants, xylan can be contained in water-soluble and water-insoluble forms [43]. Accordingly, a more water-soluble form of xylan will be more accessible to endo-1,4-β xylanases. In oats, the increase in reducing sugars was 30% at 5% substrate; at 10% of substrate, the reducing sugars decreased. This may be due to the presence of xylanase inhibitors in oats [44], which were extracted to a greater extent with 10% substrate. Considering the complex structure of plant xylan-containing products, other components of it can affect xylanase. For example, phenolic compounds and lignin derivatives, which are also contained in raw materials in varying quantities, reduce xylanase activity [45].

## 3. Materials and Methods

### 3.1. Microorganisms and Cultivation

The strain used in this work was *Nocardiopsis halotolerans* VKM Ac-2519, obtained from the All-Russian Collection of Microorganisms (http://www.vkm.ru/index.htm accessed on 27 June 2024). The bacterium was grown in a medium with the following composition (g/L): peptone—5; yeast extract—3; KH_2_PO_4_—0.2; glucose—5, pH 7.2. The bacterium was stored on a similar agar medium at a temperature of 4 °C by periodic reseeding once a month.

### 3.2. Gene Cloning and Construction of a Recombinant Producer

Genomic DNA was obtained by phenol/chloroform extraction from bacterial biomass frozen in liquid nitrogen and destroyed with a mortar and pestle. The gene encoding the enzyme was amplified without the signal peptide using PCR. Based on the sequence deposited in GenBank (NCBI Reference Sequence of proteins: WP_017571782.1), primers were designed: F: GGCTGAAGCTTACGTAGCCATCACCACCAAC; R: GGTGCTGATGGAATTCGTTCGCGCTACAGG. To amplify the DNA fragment, the following PCR conditions were used: 98 °C for 1 min; 6 cycles at 98 °C for 10 s, 55 °C for 30 s, 72 °C for 2 min; 28 cycles at 98 °C for 10 s, 72 °C for 2 min; 72 °C for 10 min. A PCR product of the appropriate size was purified from an agarose gel using the diaGene kit (Diaem, Moscow, Russia). To clone the PCR product, we used the In-Fusion CF Dry-Down PCR Cloning Kit (Clontech, Shiga, Japan), which does not require the use of restriction enzymes and ligases. The PCR product was ligated into the expression vector pPic9m treated with restriction enzymes SnaBI and EcoRI (New England Biolabs, Ipswich, MA, USA). The pPic9m vector is capable of replicating in *Escherichia coli* and *Pichia pastoris* and is derived from the pPic9K vector (Invitrogen, Waltham, MA, USA), to which a 6×His sequence was added to the C-terminus. The recombination solution was incubated at 42 °C for 30 min and then used to transform *E. coli* DH5α. The selection of clones that underwent recombination was carried out using PCR with the standard primers 5′AOX1 and 3′AOX1 for the pPic9m vector. Conditions for PCR: 95 °C for 3 min; 30 cycles of 95 °C for 40 s, 55 °C for 40 s and 72 °C for 2 min; 72 °C for 10 min. The presence of the target gene insert in the plasmid was verified by sequencing.

The recombinant plasmid containing the NhX1 gene was linearized using restriction enzyme Pme I (New England Biolabs, USA) and using electroporation (voltage 15,000 V; electrical capacity 25 F; resistance 200 mOhm; time 4.5 ms) was transferred into electrocompetent cells of *P. pastoris* GS115. *P. pastoris* GS115 colonies containing the plasmid were selected on YNB agar medium without amino acids (Sigma, Burlington, MA, USA) containing 1.34% YNB, 0.0004% biotin, and 2% glucose. To test the expression of NhX1, selected recombinant *P. pastoris* clones were grown at 29 °C for 6 days in the following medium (g/L): yeast extract—10; peptone—20%, sorbitol—10. The medium was prepared based on 100 mM K-phosphate buffer, pH 6.8. To induce NhX1, starting from the third day of cultivation, 0.5% (final concentration) methanol was added daily under sterile conditions. Protein production was monitored using SDS–PAGE electrophoresis.

### 3.3. Fermentation and Purification of NhX1

To produce the enzyme, cultivation of *P. pastoris* was carried out in a fermenter with a volume of 10 l (ANKUM-2M, Pushchino, Russia), with a working volume of 5 l. The culture of *P. pastoris* for seeding the fermenter was grown in flasks for 16 h at a temperature of 30 °C in a medium with the following composition (g/L): glucose—20, peptone—20, yeast extract—10. The culture for seeding was added to the fermenter in an amount of 10% from the working volume. For fermentation, a medium of the following composition (g/L) was used: glycerol—40; MgSO_4_ x _7_H_2_O—1; CaCL_2_—0.5; (NH_4_)_2_SO_4_—5; Na_2_HPO_4_—2; yeast extract—10; peptone—20, medium pH 5.5, cultivation temperature 30 °C. To determine the concentration of the dry biomass, *P. pastoris* cells were deposited on a Schott filter, dried at a temperature of 105 °C to a constant weight, and weighed. The protein concentration was determined by the Bradford method. The oxygen concentration during fermentation was determined using a Clark electrode, and the pH was maintained with 0.5% NaOH during cultivation. After fermentation, the *P. pastoris* biomass was removed by centrifugation at 5000× *g* 30 min. The culture liquid thus obtained was used for further purification.

The enzyme was purified from the culture liquid of *P. pastoris* using affinity chromatography on a HisTrapp FF column (GE Healthcare, Chicago, IL, USA). Purification conditions: the column was equilibrated with 20 mM Tris-HCl buffer pH 7.5 with 20 mM imidazole and 0.5 M NaCL. The protein was then eluted with the same buffer but containing 300 mM imidazole. The procedure was carried out at a speed of 1 mL/min. The resulting enzyme preparation was transferred by dialysis into 20 mM Tris–HCl buffer pH 7.5 and used in further work.

### 3.4. Characterization of NhX1

Identification of domains in the NhX1 sequence was carried out using the InterPro 86.0 program (https://www.ebi.ac.uk/interpro/ accessed on 27 June 2024) and the signal peptide was identified using the SignalP-5.0 software (http://www.cbs.dtu.dk/services/SignalP/ accessed on 27 June 2024). The search for homologous proteins was performed using BLAST (https://blast.ncbi.nlm.nih.gov/Blast.cgi accessed on 27 June 2024). Amino acid sequence alignment was performed using the CLUSTAL W algorithm [46] (https://www.ebi.ac.uk/jdispatcher/msa/muscle accessed on 27 June 2024). The phylogenetic tree was constructed using the MegaNucleotide program (https://www.megasoftware.net/active_download accessed on 27 June 2024). Hydrolytic activity toward beech xylan was determined by the formation of reducing sugars, which were detected using 3,5-dinirobenzoic acid [47]. One unit of endo-1,4-β-xylanase activity was taken to be an amount of the enzyme that catalyzes the formation per minute of reducing groups corresponding to 1 μM xylose. The studies of thermal stability, pH optimum, pH stability, temperature optimum and identification of beech xylan hydrolysis products using thin layer chromatography were carried out as described previously [22]. For experiments on the saccharification of xylan-containing raw materials, rye, barley, oats, wheat, and wheat bran were used. The raw materials were ground and a fraction of 0.2 microns in size was selected using sieves. The reaction was carried out at 40 °C in 50 mM Britton-Robinson buffer pH 6.0 for 24 h. The reaction mixture contained xylan-containing raw materials in an amount of 1%, 5% and 10% (*w*/*v*) and the enzyme in an amount of 2 U/mL, and the reaction was carried out with stirring. The control was a reaction mixture that contained the enzyme inactivated by heating for five minutes in a boiling water bath and the appropriate amount of the materials. To determine the kinetic constants, we studied the dependence of the enzymatic reaction rate on the xylan concentration. The apparent constants were calculated by nonlinear regression of data using SigmaPlot 11.0 software.

## 4. Conclusions

Thus, xylanase NhX1 from *Nocardiopsis halotolerans* effectively hydrolyzed xylan in plant xylan-containing products, which, along with its high thermal stability, indicates great potential for using the enzyme for saccharification of xylan-containing plant materials.

## Figures and Tables

**Figure 1 ijms-25-09121-f001:**
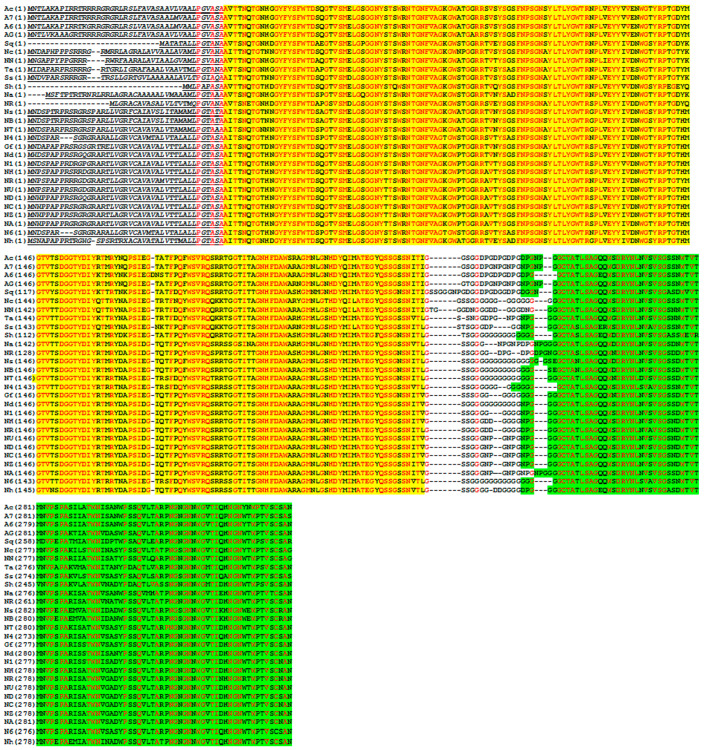
The alignment of amino acid sequences of NhX1 and its homologous proteins. Yellow—GH11 catalytic domain; Green—polysaccharide binding domain of CBM2; underlining—TAT signal peptide. Conservative amino acids are highlighted in red. Designation of strains (protein ID) is indicated in brackets: Ac—*Actinomadura catellatispora* (WP_184880026.1); A7—*Actinomadura* sp. 7K534 (WP_132050328.1); A6—*Actinomadura* sp. 6K520 (WP_131986178.1); AG—*Actinomadura* sp. GC306 (WP_131951671.1); Sq—*Salinactinospora qingdaonensis* (GAA3767778.1); Nc—*Nonomuraea composti*FMUSA5-5 (WP_168015463.1); NN—*Nonomuraea* sp. N2-4H (WXK68348.1); Ta—*Thermobifida alba* (WP_248592304.1); Ss—*Saccharothrix syringae* (WP_033431747.1); Sh—*Streptomonospora halophila* (GAA4943445.1); Na—*Nocardiopsis algeriensis* (WP_184289281.1); NR—*Nocardiopsis* sp. NRRL B-16309 (KOX10261.1); Ns—*Nocardiopsis sinuspersici* (WP_179809348.1); NB—*Nocardiopsis* sp. BMP B8015 (WP_077691514.1); NT—*Nocardiopsis* sp. TSRI0078 (WP_073702879.1); N4—*Nocardiopsis* sp. FR4 (WP_160050949.1); Gf—*Glycomyces fuscus* (PDP84525.1); Nd—*Nocardiopsis deserti* (WP_150252462.1); N1—*Nocardiopsis alborubida* (WP_061080181.1); NH—*Nocardiopsis akebiae* (WP_212643832.1); NR—*Nocardiopsis* sp. RV163 (WP_047868137.1); NU—*Nocardiopsis* sp. HUAS JQ3 (WP_274973760.1); ND—*Nocardiopsis dassonvillei* DSM 43111 (WP_013153731.1); NC—*Nocardiopsis dassonvillei* NCIM 5124 (WP_248771023.1); NZ—*Nocardiopsis dassonvillei* HZNU_N_1 (WP_019608409.1); NA—*Nocardiopsis dassonvillei* RACA_4 (WP_253794179.1); N6—*Nocardiopsis* sp. FR6 (WP_159945040.1); Nh—*Nocardiopsis halotolerans* (WP_017571782.1).

**Figure 2 ijms-25-09121-f002:**
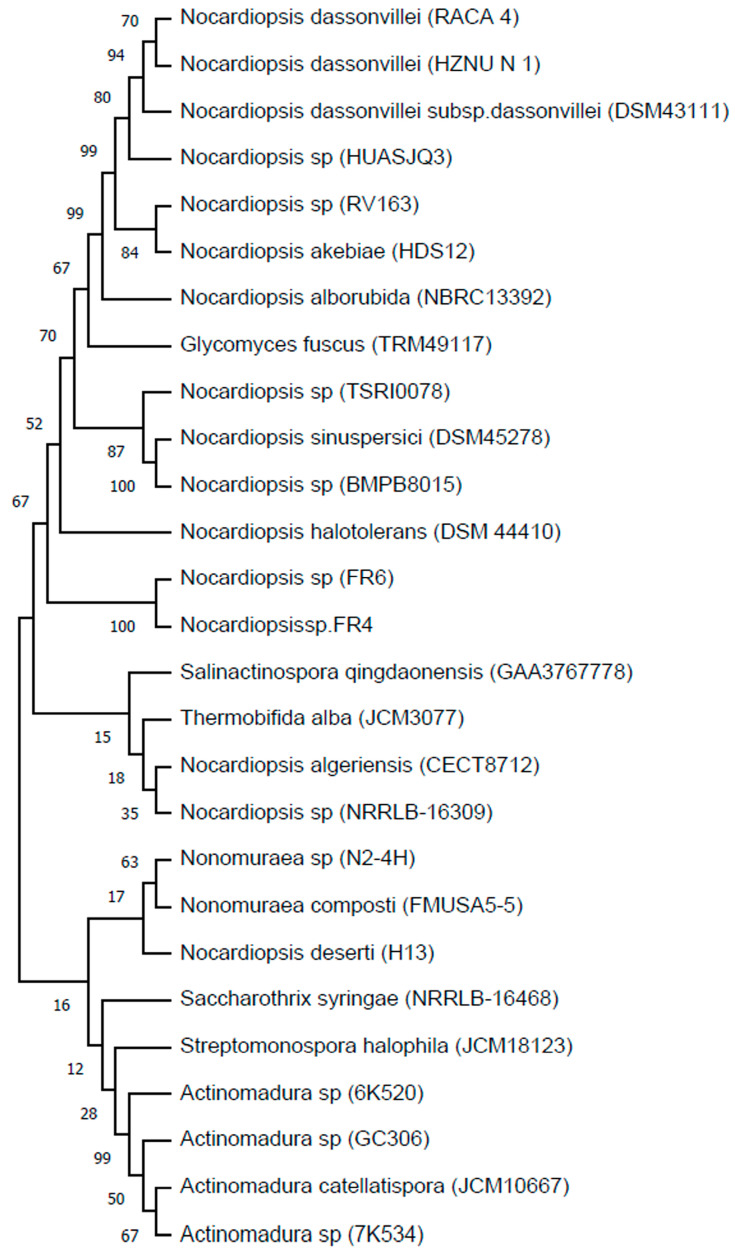
Phylogenetic tree of protein 1 of its 28 homologs from various actinobacteria. Values of the ultra-fast bootstrap analysis are presented by the numbers in the nodes.

**Figure 3 ijms-25-09121-f003:**
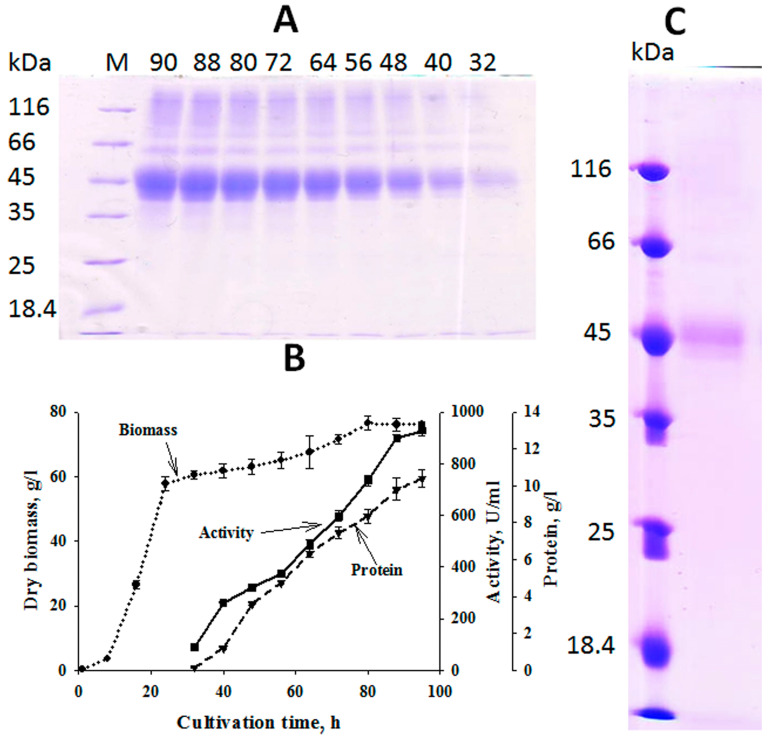
Production of endo-1,4-β-xylanase by *P. pastoris* during fermentation. (**A**) SDS–PAGE electrophoresis data. M—molecular mass markers in kDa; the numbers at the top are the cultivation time indicated in hours. (**B**) Biomass growth (dotted line); enzyme activity (solid line); protein concentration (dashed line). (**C**) SDS–PAGE electrophoresis data of the purified enzyme.

**Figure 4 ijms-25-09121-f004:**
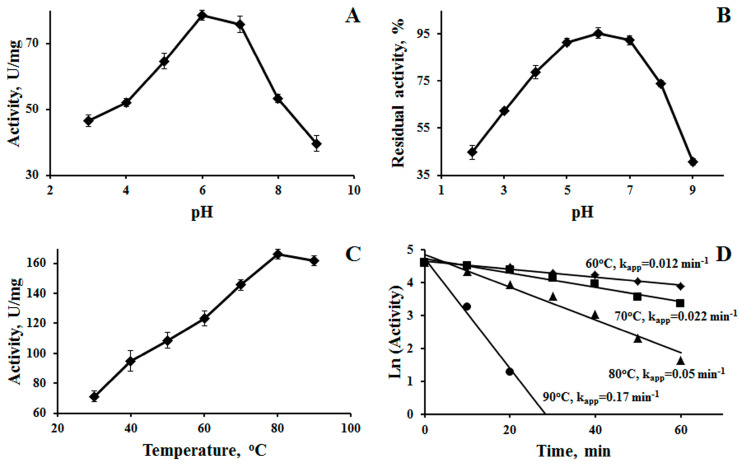
Properties of recombinant NhX1. (**A**) pH optimum, (**B**) pH stability, (**C**) temperature optimum, (**D**) thermal stability: diamonds—60 °C, squares—70 °C, triangles—80 °C, circles—90 °C.

**Figure 5 ijms-25-09121-f005:**
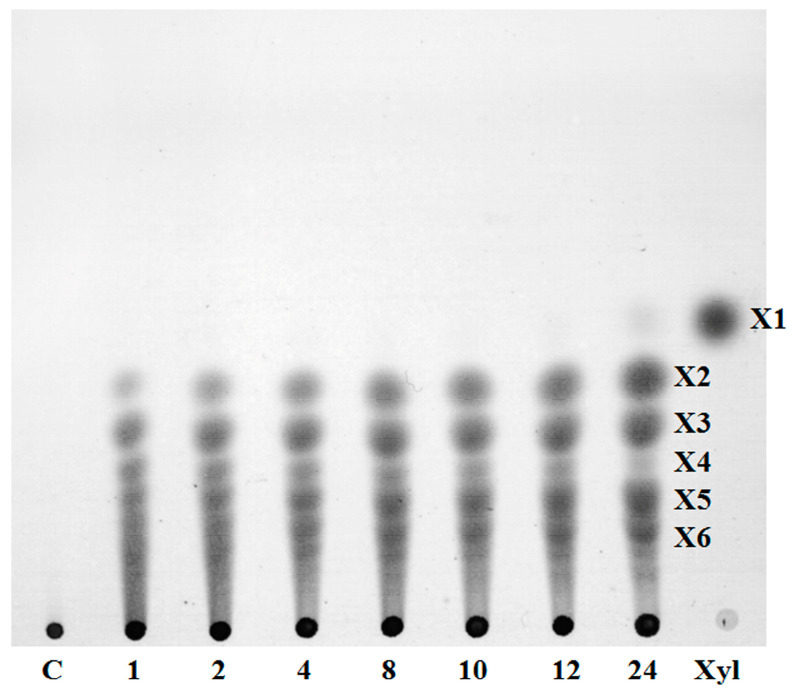
Formation of hydrolysis products from beech xylan under the action of NhX1. X1—xylose, X2—xylobiose, X3—xylotriose, X4—xylotetraose, X5—xylopentaose, X6—xylohexaose, and C—control. The numbers below are the reaction time in hours.

**Figure 6 ijms-25-09121-f006:**
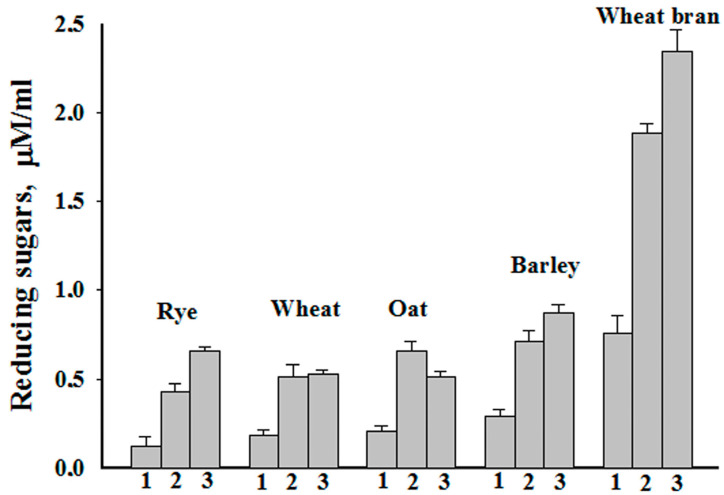
Formation of reducing sugars under the action of NhX1 at different concentrations of xylan-containing products. The results are shown minus the controls at each substrate concentration. Concentrations: 1—1%, 2—5%, and 3—10%. The bars represent the standard deviation obtained from three independent experiments.

## Data Availability

The data that support the findings of this study are available from the corresponding author, A.V.L., upon reasonable request.

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
