# Peer review of "Expression in Pichia pastoris of Thermostable Endo-1,4-β-xylanase from the Actinobacterium Nocardiopsis halotolerans: Properties and Use for Saccharification of Xylan-Containing Products"

_ijms, 2024, doi:10.3390/ijms25169121_

Round 1

Reviewer 1 Report

Comments and Suggestions for Authors

Manuscript: Expression in Pichia pastoris of thermostable endo-1,4-β-xylanase from the actinobacterium Nocardiopsis halotolerans: properties and use for saccharification of xylan-containingproducts

The manuscript by Lisov et al. demonstrated the expression of thermostable endo-1,4-β-xylanase from Nocardiopsis halotoleransin Pichia pastorisand its application Overall, the manuscript is interesting and requires revision as follows:

Comments

1.    Title. Please delete “.”

2.    Abstract, the presentation can be revised with issues that to be solved, present study/finding, novelty and perspectives.

3.    Introduction, the biomass consists of cellulose, hemicelluloses (including xylan) and lignin with a variation in these compositions depending on the type of biomass. The sachharification of xylan using xylanase alone cannot be done/feasible (per the application in this study). In addition, the hydrolysis of biomass led to generation of phenolics that interfere in microbial biotransformation of biomass hydrolysate to useful products i.e. https://doi.org/10.1016/j.enzmictec.2023.110301. Therefore, the author’s should add such information.

4.    The kinetic parameters data of recombinant enzyme should be provided, and compared with literature for significance.

5.    Figure 4, instead of relative activity (%), the actual activity should be provided in terms of specific activity.

6.    How about t1/2 value at different temperatures?

7.    Figure 6, the composition of reducing sugars should be analyzed along with phenolics.

8.    Overall discussion is weak. It can be improved along with recent updates.

Comments on the Quality of English Language

Minor typos 

Author Response

We thank the reviewer for a job well done in improving the quality of the manuscript.

  1. Title. Please delete “.”

- deleted “.”

  1. Abstract, the presentation can be revised with issues that to be solved, present study/finding, novelty and perspectives.

- The abstract briefly describes the results obtained during the work. Unfortunately, according to the journal's rules, the size of the abstract is limited. Therefore, it is difficult for us to add anything.

  1. Introduction, the biomass consists of cellulose, hemicelluloses (including xylan) and lignin with a variation in these compositions depending on the type of biomass. The sachharification of xylan using xylanase alone cannot be done/feasible (per the application in this study). In addition, the hydrolysis of biomass led to generation of phenolics that interfere in microbial biotransformation of biomass hydrolysate to useful products i.e. https://doi.org/10.1016/j.enzmictec.2023.110301. Therefore, the author’s should add such information.

- We agree that plant biomass is a complex material consisting of various chemical compounds. In our article, we will primarily focus on the transformation of xylan. We will discuss the problem associated with the complexity of plant biomass in the Discussion. Including the mentioned article.

  1. The kinetic parameters data of recombinant enzyme should be provided, and compared with literature for significance.

- We determined the kinetic constants of the enzyme for beechwood xylan and discussed this point.

  1. Figure 4, instead of relative activity (%), the actual activity should be provided in terms of specific activity.

- replaced residual activity with specific activity in Fig 4 A,C

  1. How about t1/2 value at different temperatures?

- we added data on the enzyme half-inactivation time to the text. To improve the presentation of the results of the thermal inactivation experiments, the data at Fig.4C are presented in semi-logarithmic coordinates.

  1. Figure 6, the composition of reducing sugars should be analyzed along with phenolics.

- We agree that various components of plant materials may influence the efficiency of saccharification by xylanase. But in this work, we studied the effect of xylanase only on the total release of reducing sugars without taking into account the influence of other components, including phenols. We have included a discussion of this issue in the article. We plan to study this issue in our future works.

  1. Overall discussion is weak. It can be improved along with recent updates.

- Thank you for your comment. We expanded the discussion and tried to take into account the reviewer's wishes.

Reviewer 2 Report

Comments and Suggestions for Authors

The topic is of scientific significance.

However, there are a few queries.

Are the values statistically significantly  higher than the control

i.e. The formation of reducing sugars.

Was this formation of reducing sugars influenced by the substrate concentrations?

The need is to prove the significance of the work by conducting animal feed experiments and evaluating their influence on animal body weight or any other metabolic parameter, which is indicative of health.

Author Response

We thank the reviewer for a job well done in improving the quality of the manuscript.

Are the values statistically significantly higher than the control? i.e. The formation of reducing sugars.

- We remade the experiment on the formation of reducing sugars under the action of xylanase in accordance with the reviewer's requirements and carried out statistical processing of the results. In Figure 6, results are shown minus the control. Therefore, the data on the formation of reducing sugars is statistically reliable.

Was this formation of reducing sugars influenced by the substrate concentrations?

- We studied the dependence of the efficiency of saccharification in chavisibility at substrate concentrations of 1, 5, 10%. The results are presented in the manuscript.

The need is to prove the significance of the work by conducting animal feed experiments and evaluating their influence on animal body weight or any other metabolic parameter, which is indicative of health.

- Such research is certainly of interest. We are planning such a study. However, this is a separate voluminous work that deserves a separate publication.

Reviewer 3 Report

Comments and Suggestions for Authors

Expression in Pichia pastoris of thermostable endo-1,4-β-xylanase from the actinobacterium Nocardiopsis halotolerans: properties and use for saccharification of xylan-containing products.

Introduction: Endo-1,4-β-xylanase is an industrially important enzyme, obtained from various producers, both natural and recombinant. Thermostable endo-1,4-β-xylanases are obtained primarily from thermophilic microorganisms, as well as via protein engineering methods.

The last paragraph of the introduction spells out the aims: “This work describes the preparation and study of the properties of endo-1,4-β-xylanase from the actinobacterium Nocardiopsis halotolerans. N. halotolerans was isolated from saline soils [14]. A complete genome has been determined for this bacterium and deposited in GenBank. The N. halotolerans genome contains genes encoding glycosyl hydrolases, including endo-1,4-β-xylanases. The goal of the work was to obtain a preparation of re-combinant endo-1,4-β-xylanase from the actinobacterium N. halotolerans, to study its properties and biotechnological potential.

The introduction does not mention Pichia pastoris as the final expression species? One assumes that this species grows at a higher rate than actinobacteria and would be the method chosen for commercial synthesis?

Materials and methods:

3.1. Microorganisms and cultivation: This is fine.

3.2. Gene cloning and construction of a recombinant producer. Your section heading states 3.3 rather than 3.2; please correct. Otherwise, the section is fine.

3.3. Fermentation and purification of NhX1: This is OK except for last sentence (line 263) where it states “The resulting drug was transferred by dialysis into 20 mM Tris-HCl buffer pH 7.5 and used in a further work”. I don’t understand the use of the term “drug”; which drug? Is this an error?

3.4. Characterization of NhX1: BLAST search and CLUSTAL alignment followed by phylogenetic tree for sequences. These all seem appropriate except for enzyme units. Beech xylan used as substrate. Enzyme units are amount of enzyme that can form xylose equivalents of reducing groups, i.e. 1 μM xylose per minute. Surely the units should be micromoles per minute (an amount) rather than micromolar per minute (a concentration)? If the latter, you must define the volume of your assay mixture since the same unit of enzyme would produce a different concentration if it was in 0.1, 1 or 2ml assay. Slightly more details required in methods? How was biomass harvested from the culture? Centrifugation or filtration? How was enzyme extracted from cells? Were cells sonicated or lysed by chemicals or enzymes, or did you use high pressure cell press?

Results and discussion:

2.1. Characteristics of the amino acid sequence and phylogenetic analysis: Reveals many homologous proteins. The homologous proteins have not been previously studied, so cannot say if properties are similar to endo-1,4-β-xylanase from N. halotolerans. Figure 1 shows alignments for 27 such homologs. Figure 2 shows phylogenetic tree of homologs.

2.2. Cloning, expression and purification: Authors produce a high concentration of biomass rich in target protein by growth in fed batch culture using glycerol as main C/E substrate, and methanol as inducer when glycerol is depleted. After 90 hours, cells are harvested. Figure 3 shows data from production over time for target protein (biomass, activity and amount of protein using assays and SDS PAGE electrophoresis).

2.3. PropertiesofNhX1

Following purification to homogeneity (figure 3C) the enzyme protein properties were studied. The enzyme hydrolysed xylan but showed no activity against cellulose, CM-cellulose, beta-glucan, lichenin, and starch. The optimal pH value for xylan hydrolysis was between pH 6.0 – 7.0 shown in figure 4A. The enzyme exhibited high thermal stability and high catalytic activity at 60-80°C. Interesting discussion about stability and products of xylan hydrolysis (xylooligosaccharides). Figure 6 shows formation of reducing sugars from different xylan sources (rye; wheat, oats, barley and wheat bran).

4. Conclusions: Authors state “Thus, xylanase NhX1 from Nocardiopsis halotolerans effectively hydrolyzed xylan of plant xylan-containing products, which, along with high thermal stability, indicates a great potential for using the enzyme to improve the nutritional properties of animal feed”.

All together this is an interesting paper with only a few errors (pointed out)

Comments on the Quality of English Language

Quality of English is good. Only a few minor errors, pointed out in the authors review.

Author Response

We thank the reviewer for his good work in helping us improve our text.

The introduction does not mention Pichia pastoris as the final expression species? One assumes that this species grows at a higher rate than actinobacteria and would be the method chosen for commercial synthesis?

  • Yes, Pichia pastoris was chosen because it is used for the industrial production of enzymes. We have added a mention of Pichia pastoris in the introduction.

3.2. Gene cloning and construction of a recombinant producer. Your section heading states 3.3 rather than 3.2; please correct. Otherwise, the section is fine.

  • replaced 3.3 with 3.2

3.3. Fermentation and purification of NhX1: This is OK except for last sentence (line 263) where it states “The resulting drug was transferred by dialysis into 20 mM Tris-HCl buffer pH 7.5 and used in a further work”. I don’t understand the use of the term “drug”; which drug? Is this an error?

  • replaced the “drug” with the enzyme preparation

3.4. Characterization of NhX1: BLAST search and CLUSTAL alignment followed by phylogenetic tree for sequences. These all seem appropriate except for enzyme units. Beech xylan used as substrate. Enzyme units are amount of enzyme that can form xylose equivalents of reducing groups, i.e. 1 μM xylose per minute. Surely the units should be micromoles per minute (an amount) rather than micromolar per minute (a concentration)? If the latter, you must define the volume of your assay mixture since the same unit of enzyme would produce a different concentration if it was in 0.1, 1 or 2ml assay. Slightly more details required in methods? How was biomass harvested from the culture? Centrifugation or filtration? How was enzyme extracted from cells? Were cells sonicated or lysed by chemicals or enzymes, or did you use high pressure cell press?

  • In our work, we used international units of enzyme activity. We are talking about the amount of substance, volume was taken into account when calculating activity units. The enzyme produced during fermentation was extracellular. We added information about biomass removal to the text.

Round 2

Reviewer 1 Report

Comments and Suggestions for Authors

The manuscript has been revised well. 

Comments

1. In lines 23 and 23, the SD value can be deleted. 

Author Response

We thank the reviewer for his work.

We respond to the comment raised.

1. In lines 23 and 23, the SD value can be deleted. 

- We deleted SD values.

Reviewer 2 Report

Comments and Suggestions for Authors

Validation studies are needed in the future.

Author Response

We thank the reviewer for his work.

Reviewer 3

Validation studies are needed in the future.

- We agree that validation experiments are necessary. We plan to hold them in the future.